# SARS-CoV-2 induces a durable and antigen specific humoral immunity after asymptomatic to mild COVID-19 infection

Sebastian Havervall[1], August Jernbom Falk[2], Jonas Klingström[3,4], Henry Ng[5], Nina Greilert-Norin[1], Lena Gabrielsson[1], Ann-Christin Salomonsson[1], Eva Isaksson[1], Ann-Sofie Rudberg[8], Cecilia Hellström[2], Eni Andersson[2], Jennie Olofsson[2], Lovisa Skoglund[2], Jamil Yousef[2], Elisa Pin[2], Wanda Christ[3], Mikaela Olausson[4], My Hedhammar[6], Hanna Tegel[6], Sara Mangsbo[7], Mia Phillipson[5], Anna Månberg[2], Sophia Hober[6], Peter Nilsson[2], Charlotte Thålin[1]*

1 Department of Clinical Sciences, Karolinska Institute, Danderyd Hospital, Stockholm, Sweden, 2 Division of Affinity Proteomics, Department of Protein Science, KTH Royal Institute of Technology, SciLifeLab, Stockholm, Sweden, 3 Centre for Infectious Medicine, Department of Medicine Huddinge, Karolinska Institute, Stockholm, Sweden, 4 Department of Microbiology, Public Health Agency of Sweden, Solna, Sweden, 5 Department of Medical Cell Biology, Uppsala University, SciLifeLab, Uppsala, Sweden, 6 Division of Protein Technology, Department of Protein Science, KTH Royal Institute of Technology, Stockholm, Sweden, 7 Department of Pharmaceutical Biosciences, Science for Life Laboratory, Uppsala University, Uppsala, Sweden, 8 Department of Neurology, Danderyd Hospital, Stockholm, Sweden

☯ These authors contributed equally to this work.
* charlotte.thalin@ki.se

**Data Availability Statement:** Data cannot be shared publicly as it contains sensitive personal information which is protected by the GDPR. Data

## Abstract

Current SARS-CoV-2 serological assays generate discrepant results, and the longitudinal characteristics of antibodies targeting various antigens after asymptomatic to mild COVID-19 are yet to be established. This longitudinal cohort study including 1965 healthcare workers, of which 381 participants exhibited antibodies against the SARS-CoV-2 spike antigen at study inclusion, reveal that these antibodies remain detectable in most participants, 96%, at least four months post infection, despite having had no or mild symptoms. Virus neutralization capacity was confirmed by microneutralization assay in 91% of study participants at least four months post infection. Contrary to antibodies targeting the spike protein, antibodies against the nucleocapsid protein were only detected in 80% of previously anti-nucleocapsid IgG positive healthcare workers. Both anti-spike and anti-nucleocapsid IgG levels were significantly higher in previously hospitalized COVID-19 patients four months post infection than in healthcare workers four months post infection (p = 2*10^{-23} and 2*10^{-13} respectively). Although the magnitude of humoral response was associated with disease severity, our findings support a durable and functional humoral response after SARS-CoV-2 infection even after no or mild symptoms. We further demonstrate differences in antibody kinetics depending on the antigen, arguing against the use of the nucleocapsid protein as target antigen in population-based SARS-CoV-2 serological surveys.

are available on request from the SciLifeLab Data Repository (doi: 10.17044/scilifelab.13567355.v2) for researchers who meet the criteria for access to sensitive personal data.

**Funding:** This study was funded by Region Stockholm (CT, SoH), Knut and Alice Wallenberg foundation (CT, SoH), Jonas & Christina af Jochnick foundation (CT), Lundblad family foundation (CT), Science for Life Laboratory (PN), Erling-Persson family foundation (SoH), Svenska Sällskapet för Medicinsk Forskning (SM), and Swedish Research Council (JK), CIMED (JK). The funders had no role in study design, data collection and analysis, decision to publish, or preparation of the manuscript.

**Competing interests:** The authors have declared that no competing interests exist.

## Introduction

Severe acute respiratory syndrome coronavirus 2 (SARS-CoV-2), causing the coronavirus disease 2019 (COVID-19), has taken global pandemic proportions. Despite a steady increase in the number of fatalities worldwide, the vast majority of infected individuals develop no or mild symptoms. The rapid spread of SARS-CoV-2 is likely facilitated by a substantial portion of asymptomatic and pre-symptomatic transmission [1, 2]. Understanding the durability and functionality of the immune response in mild and asymptomatic cases is therefore critical in the attempt to contain the disease and to gain insight of the potential of re-infection.

Antibodies targeting various virus-encoded proteins are central players in conveying protective immunity against viral infections such as SARS-CoV-2. The most commonly targeted antigens in currently available serology assays are the SARS-CoV-2 spike glycoprotein, which enables viral access to the host cell, and the abundant and highly conserved nucleocapsid protein [3, 4]. The SARS-CoV-2 spike glycoprotein is the main target antigen for neutralizing antibodies and vaccine development. It is now well-established that circulating IgG antibodies to SARS-CoV-2 are detected in the majority of infected individuals after 9–21 days from symptom onset [5–9]. The kinetics, duration and efficacy of circulating SARS-CoV-2 antibodies are however, due to the novelty of the virus, less established. In fact, conflicting studies provide data on a rapid decline in circulating IgG antibodies within weeks after COVID-19 [10–12], especially after mild disease [13–15], while others report detectable antibodies up to two to six months after symptom onset [16–19].

The COMMUNITY (COVID-19 Immunity) study is an ongoing longitudinal study investigating long-term immunity after COVID-19 in a large group of individuals with a wide variety of COVID-19 symptoms. Between April 15th and May 8th, 2020, 2149 health care workers (HCW), and 118 hospitalized COVID-19 patients were included in the study. Cross-sectional clinical, demographic and serological data of the HCW at study inclusion has been presented elsewhere [20]. Briefly, the cohort comprised 85% women (1815/2149) with a mean age of 44 (SD 12). 410 HCW presented antibodies recognizing both spike and nucleocapsid at inclusion, among which 87% (357/410) reported mild or asymptomatic infection. The objective of this first follow-up was to assess the duration and efficacy of circulating antibodies four months after infection. Our findings show that a durable anti-spike IgG response is generated in the vast majority of convalescent individuals, even after asymptomatic or mild infection, and that these antibodies remain capable of virus neutralization. In contrast, anti-nucleocapsid IgG levels declined in a large portion of individuals with asymptomatic or mild disease, but not in individuals with severe to critical disease. These findings have important implications for public health planning, as well as for assessing potential risk of reinfection and serological evaluation of vaccine responses.

## Results

### Characteristics of study participants

A total of 1965 HCW and 59 convalescent COVID-19 patients remained in the study for the four-month follow-up between August 24th and September 11th, 2020. IgG antibodies against the SARS-CoV-2 spike protein (full length trimer) and a 118 aa-long C-terminal domain of the nucleocapsid protein were analyzed in all samples at inclusion in April/May 2020 and at four-month follow-up. The majority of HCW were women (n = 1669, 85%) and the mean age was 44 (SD 12) years. Of the 1965 HCW, 350 individuals (18%) presented antibodies recognizing both spike and nucleocapsid at inclusion, whereas 31 (2%) presented only anti-spike IgG and 80 (4%) presented only anti-nucleocapsid IgG. Of the 461 HCW presenting antibodies

recognizing spike and/or nucleocapsid at inclusion, 406 (88%), had mild symptoms prior to study inclusion, and 55 (12%) had been asymptomatic. The patient group was predominantly male (69%), and the mean age was higher than that of HCW; 56 (SD 14) years ($p = 3^*10^{-9}$). All COVID-19 patients were shown to have antibodies recognizing both spike and nucleocapsid at study inclusion.

### Persistence of circulating anti-spike IgG

At the four-month follow-up, we first assessed the longevity of anti-spike IgG. The vast majority of HCW who were anti-spike IgG positive at inclusion (96%; 366/381), and all convalescent COVID-19 patients remained anti-spike IgG positive at the four-month follow-up. Follow-up levels of anti-spike IgG were significantly higher in convalescent COVID-19 patients (21238 MFI [AU] (IQR 19371–22737)) than HCW (12010 MFI [AU] (IQR 7747–16629)); $p = 2^*10^{-23}$, Fig 1A). In addition, high antibody levels in HCW were associated with several self-reported symptoms prior to study inclusion, including fever, dyspnea, cough, abdominal pain, ageusia, malaise, and anosmia (Fig 1B).

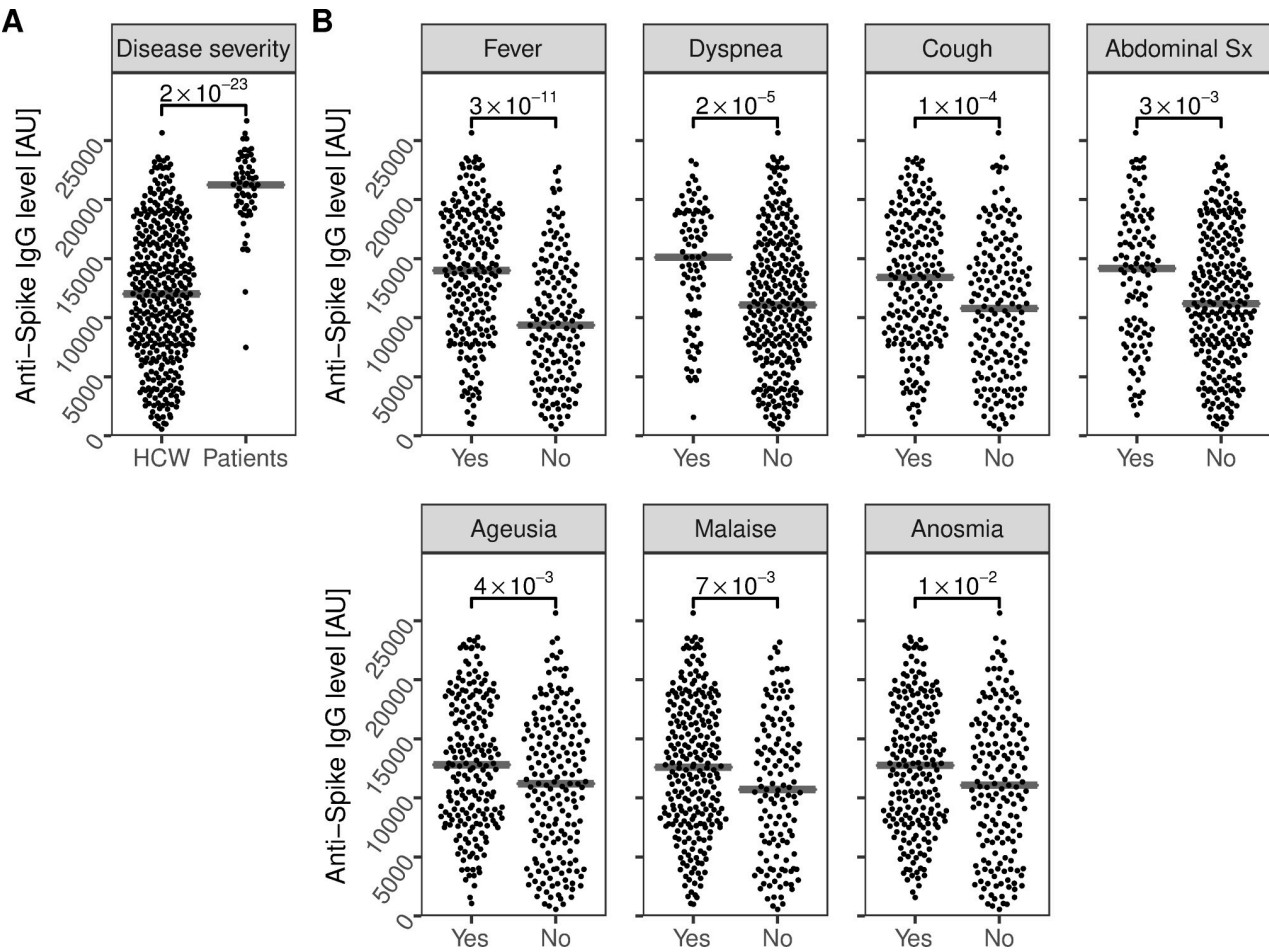

**Fig 1. Four-month follow-up levels of anti-Spike IgG are associated to disease severity and COVID-19 symptoms.** A) Four-month follow-up levels of anti-spike IgG were significantly higher in convalescent COVID-19 patients compared to HCW. B) In addition, four-month follow-up levels were significantly increased in HCW with self-reported fever, dyspnea, cough, abdominal symptoms, malaise, anosmia, or ageusia prior to study inclusion. Crossbars depict the median. P-values are shown with brackets. Sx; symptoms. AU: Arbitrary Units.

## Persistence of circulating anti-nucleocapsid IgG

Although nucleocapsid antibodies are not believed to be capable of direct neutralization of the SARS-CoV-2 virus, the SARS-CoV-2 nucleocapsid antigen remains target in many commercially available assays due to its high abundance and immunogenicity [21]. The nucleocapsid has furthermore reported a higher sensitivity compared to the spike protein when screening populations early in the seroconversion phase [22]. We therefore proceeded to analyze anti-nucleocapsid IgG at the four-month follow-up. Of the 59 convalescent COVID-19 patients, 58 (98%) remained seropositive for anti-nucleocapsid IgG at the four-month follow-up. In contrast to anti-spike IgG, only 80% (342/430) of previously anti-nucleocapsid IgG positive HCW remained anti-nucleocapsid IgG positive at four-month follow-up. Still, the four-month follow-up levels were significantly lower in HCW (8584 MFI [AU] (IQR 4396–12693)) than in convalescent COVID-19 patients (14966 MFI [AU] (IQR 11744–16499); p = $2^*10^{-13}$)) (Fig 2A). The four-month follow-up levels of anti-nucleocapsid IgG among HCW were, in line with follow-up levels of anti-spike IgG, associated with several self-reported symptoms prior to study inclusion (Fig 2B).

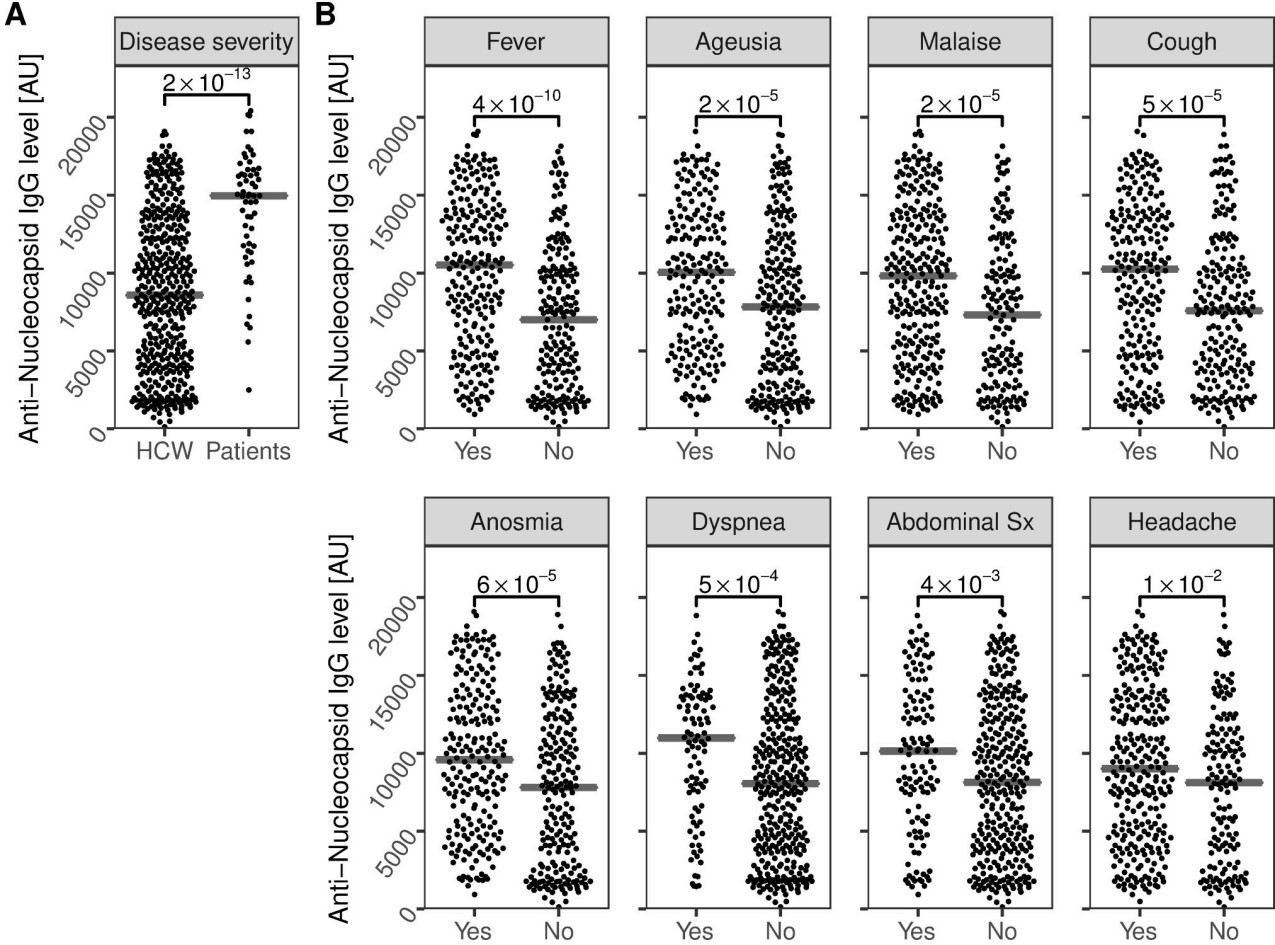

**Fig 2. Four-month follow-up levels of anti-nucleocapsid IgG are associated to disease severity and to COVID-19 symptoms.** A) Four-month follow-up levels of anti-nucleocapsid IgG were significantly higher in convalescent patients compared to HCW. B) In addition, four-month follow-up levels were significantly increased in HCW with self-reported fever, ageusia, malaise, cough, anosmia, dyspnea, abdominal symptoms, or headache prior to study inclusion. Crossbars depict the median. P-values are shown with brackets. Sx; symptoms. AU: Arbitrary Units.

## SARS-CoV-2 neutralizing antibodies

Using a microneutralization assay, SARS-CoV-2 neutralizing potential was determined in all 425 individuals who were anti-spike IgG positive both at inclusion and at four-month follow-up (366 HCW and 59 convalescent COVID-19 patients) and in a subgroup of anti-spike IgG negative individuals at four-month follow-up (197 HCW). Neutralizing potential was confirmed in 94% (401 of 425) of anti-spike IgG positive samples and in none of the anti-spike IgG negative samples (Fig 3A). Anti-spike IgG antibodies were observed at a wide range of

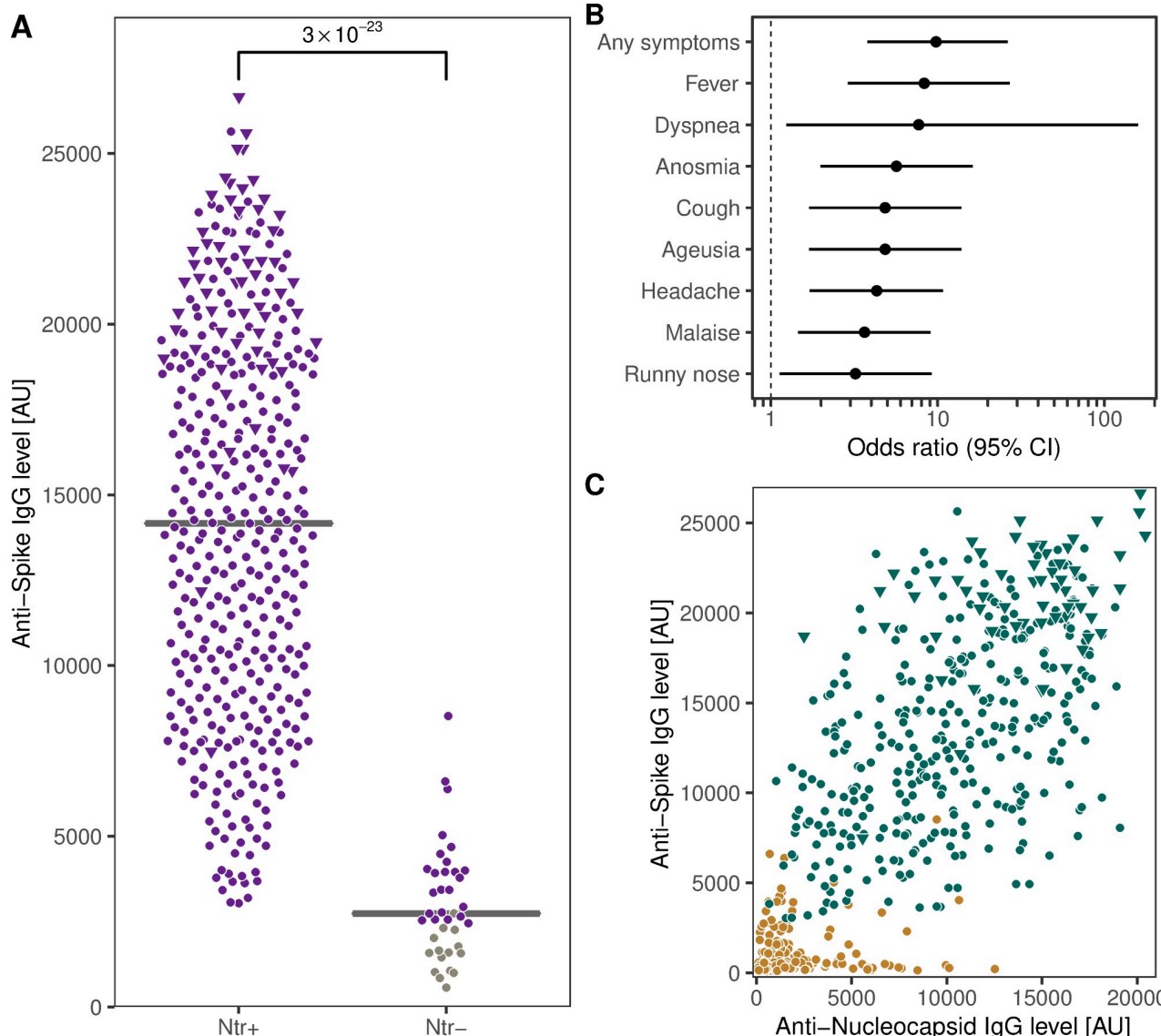

**Fig 3. Virus neutralization was confirmed in the vast majority of anti-spike IgG positive samples, and associated to COVID-19 symptoms.** A) Four-month follow-up anti-spike IgG levels in HCW or convalescent COVID-19 patients who were anti-spike IgG positive at study inclusion were significantly higher in serum from individuals with SARS-CoV-2 neutralizing potential compared to non-neutralizing samples. Purple: Anti-spike IgG positive individuals at four-month follow-up. Grey: Anti-spike IgG negative individuals at four-month follow-up. B) SARS-CoV-2 neutralizing potential of HCW who were anti-spike IgG positive at both study inclusion and follow-up was significantly associated with COVID-19 symptoms prior to study inclusion, shown with odds ratios of neutralization potential for individually self-reported symptoms C) Neutralization potential was not found in samples with high levels of anti-nucleocapsid IgG alone. Green: SARS-CoV-2 neutralizing potential. Brown: No SARS-CoV-2 neutralizing potential. Circles (panel A and C): HCW. Triangles: Convalescent COVID-19 patients. AU: Arbitrary Units. CI: Confidence Interval.

levels, and also samples with low levels were capable of virus neutralization (Fig 3A). Interestingly, virus neutralization capacity in HCW who were anti-spike IgG positive both at study inclusion and follow-up was found to be associated with COVID-19 symptoms prior to study inclusion, with an almost ten-fold probability (OR = 9.8 (95% CI 3.8–26)) of virus neutralization capacity if symptomatic compared to asymptomatic (Fig 3B). As expected, no virus neutralization potential was found in any samples with anti-nucleocapsid IgG alone (Fig 3C). The marked association of virus neutralization with anti-spike IgG, and the lack thereof with anti-nucleocapsid IgG, was confirmed using a multivariable logistic regression model accounting for the interaction of the IgGs (pseudo-$R^2$ = 0.95; $OR_{0\text{-}5000\ AU}$ (CI): anti-spike = 521 (45–12000), anti-nucleocapsid = 5.9 (0.62–35)).

## Seroconversion during the study period

Among HCW who were anti-spike IgG negative (n = 1584) or anti-nucleocapsid IgG negative (n = 1535) at study inclusion, 8% (134/1584) developed anti-spike IgG, and 7% (113/1535) developed anti-nucleocapsid IgG, respectively, during the four-month follow-up period. Seroconversion was associated with lower age for both antigens (mean (SD) years: anti-spike IgG: seroconversion = 42 (11), seronegative = 44 (12), p = 0.01; anti-nucleocapsid IgG: seroconversion = 41 (11), seronegative = 44 (12), p = 0.003), but not with sex (anti-spike IgG: $OR_{male}$ = 0.98 (0.58–1.6), p = 1; anti-nucleocapsid IgG: $OR_{male}$ = 1.2 (0.72–2), p = 0.4). Similar to seroconversion prior to study inclusion [20], seroconversion during the four-month follow-up was for both antigens associated with symptoms compatible with COVID-19 (Fig 4).

## Discussion

Understanding the long-term humoral response including virus neutralization capacity in asymptomatic to mild SARS-CoV-2 infections is key in estimating the immunity on a population basis, potential risk of reinfection and vaccine responses. In this longitudinal study including a large group of individuals with a wide range of COVID-19 symptoms, we show that the vast majority remain seropositive for SARS-CoV-2 anti-spike IgG at least four months post infection. We furthermore corroborate prior findings of strong concordance between SARS-CoV-2 anti-spike IgG and virus neutralization capacity, supporting a long-lasting and durable immunity after COVID-19 infection also in individuals with no or mild symptoms. Anti-nucleocapsid IgG, however, declined in individuals with asymptomatic to mild COVID-19 disease, implying that this antigen may not be a useful target in long-term serological population studies.

Our findings of a durable IgG response are consistent with recent studies showing stable antibody levels for up to 2–6 months [16–19]. Several other studies, however, report a rapid decline in circulating SARS-CoV-2 antibodies [10–12], especially after mild disease [13–15]. Since the onset of the COVID-19 pandemic a plethora of serological assays have emerged, using different methods such as ELISA, CLIA, lateral flow and multiplex systems [23]. Although the discrepancies regarding the longevity of SARS-CoV-2 antibodies may well stem from variations in sensitivity and specificity of these assays, the target antigen of choice is a likely contributing factor. Notably, many of the widely used commercially available serological assays target the SARS-CoV-2 nucleocapsid antigen or linear peptides of the protein [23, 24]. Early investigations [25, 26] presented results indicating that detection of antibodies against the nucleocapsid protein render more sensitive analyses than detection of antibodies against the spike protein, and several assays targeting the nucleocapsid protein have been validated to high sensitivities and specificities [23]. These validations were, however, conducted on samples taken acutely or shortly after infection where the nucleocapsid protein has shown to provide

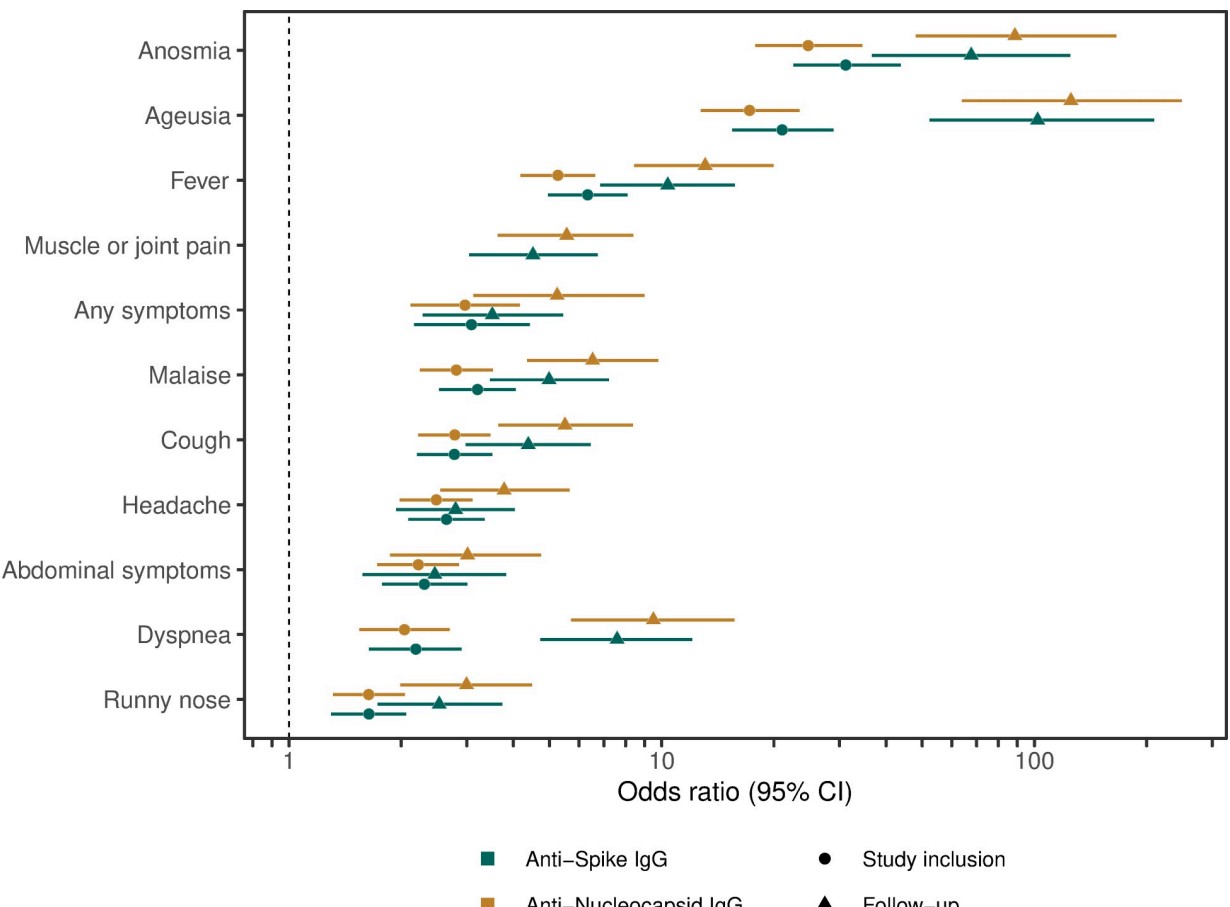

**Fig 4. Seroconversion was associated with prior COVID-19 symptoms.** Seroconversion to anti-spike IgG (green) and anti-nucleocapsid IgG (brown) prior to study inclusion (circles) and during the follow-up period (triangles) was associated with self-reported anosmia, ageusia, fever, muscle or joint pain, presence of any symptoms, malaise, cough, headache, abdominal symptoms, dyspnea, or runny nose. CI: Confidence Interval.

high sensitivity [22]. The different patterns of anti-spike IgG and anti-nucleocapsid IgG responses during the acute and convalescent phase in this study imposes questions regarding accuracy of these assays over time and emphasizes the importance of careful antigen selection. Our findings imply that serology assays aiming at assessing long-term immunity should be based on the spike protein rather than on the nucleocapsid protein.

It is well documented that SARS-CoV-2 antibody levels correlate to COVID-19 disease severity [10, 13, 27, 28]. Duration and levels of both anti-spike IgG and anti-nucleocapsid IgG at four-month follow-up were higher in convalescent COVID-19 patients, suffering severe to critical COVID-19, than in seropositive HCW with no or mild symptoms, supporting an association to disease severity. In addition, both IgG levels and virus neutralizing potential were associated with self-reported symptoms compatible with COVID-19 infection prior to study inclusion in the HCW group. However, the HCW group was both younger and presumably healthier than the convalescent COVID-19 patient group. The HCW group furthermore comprised 85% women, whereas the convalescent COVID-19 patient group comprised only 31% women, hampering comparisons between the groups.

A portion of HCW that were seronegative at inclusion seroconverted during the four-month follow-up. In line with data presented from April/May 2020 [20], we now further

corroborate associations between seroconversion and certain COVID-19 symptoms. As shown in April/May, the strongest associations remained to anosmia and ageusia, emphasizing that these symptoms should be included in routine screening guidance.

Although this study is strengthened by the large sample size and the longitudinal design with close to complete follow-up, there are certain limitations worth noting. Study inclusion took place simultaneously for HCW and hospitalized COVID-19 patients, regardless of whether and when HCW had symptoms compatible with COVID-19. A relatively large portion of seropositive HCW furthermore reported to have had no symptoms. Although the time window of infection is quite narrow in this group, considering that study inclusion started early in the Swedish pandemic, the precise time of infection remains uncertain. A direct comparison of antibody levels between HCW and hospitalized COVID-19 patients at study inclusion was therefore not appropriate. However, the levels and efficacy of neutralizing antibodies at follow-up, which was within 4–5 months post infection in both the patient and HCW groups, are more indicative of a persistent measurable humoral immunity than the dynamics between initial sampling and follow-up.

Taken together, our findings imply a strong and long-lasting humoral immune response against SARS-CoV-2, even after asymptomatic or mild infection. We furthermore reveal different patterns of acute and convalescent anti-spike IgG and anti-nucleocapsid IgG responses, and show that anti-spike IgG remain detectable in 96% of individuals while anti-nucleocapsid IgG declined to undetectable levels in 20% of the study group with mild infection. These findings have high relevance in gaining understanding of long-term humoral immunity after SARS-CoV-2 infection, and provide important insights towards public health planning, potential risk of reinfection and evaluation of long-term vaccine responses.

## Methods

### Study population

The longitudinal COMMUNITY study (COVID-19 Biomarker and Immunity study, dnr 2020–01653) is conducted at Danderyd Hospital, Stockholm, Sweden. The study population and hospital setting are described elsewhere [20]. A total of 2149 HCW and 118 hospitalized COVID-19 patients were included at baseline [20]. COVID-19 patients were diagnosed by reverse-transcriptase PCR viral detection of oropharyngeal or nasopharyngeal swabs, and the only exclusion criterium was age <18 years. PCR viral detection was not available for HCW, regardless of symptoms, prior to study inclusion. HCW were eligible to participate in the study irrespective of whether they had had symptoms since the COVID-19 outbreak onset or not. All study participants (HCW and convalescent COVID-19 patients) were invited for a follow-up visit between August 25[th] and September 17[th], 2020. 91% (1969/2149) of HCW and 50% (59/118) of convalescent COVID-19 patients came for the follow-up. HCW not hospitalized due to COVID-19 before study inclusion (n = 1965) were included in this study. Convalescent COVID-19 patients who did not come for the follow-up were either diseased (n = 14) or did not answer on repeated invitations (n = 45). Demographic data was obtained from medical journals. A questionnaire was completed by all HCW prior to each blood sampling, comprising demographics (age and sex), self-reported predefined symptoms compatible with COVID-19 (fever, headache, anosmia, ageusia, cough, malaise, common cold, abdominal pain, sore throat, shortness of breath, joint/muscle pain) prior to blood sampling, occupation, work location and self-reported exposure to patients or household members with confirmed COVID-19 infection. 100% of HCW completed the follow-up questionnaire.

The study was approved by the Swedish Ethical Review Authority (dnr 2020–01653), and written informed consent was obtained from all health care workers. Due to risk of contagion,

written informed consent was not obtained from the patients, and oral informed consent was obtained instead, or in the case of incapacity, from their next of kin. The Swedish Ethical Review Authority approved use of oral consent and the oral consent was documented in the patient's medical record and in a separate file kept with the responsible researcher. All methods were carried out in accordance with relevant guidelines and regulations.

## Serological analyses of antibodies

Venous blood samples were obtained at study inclusion and at the four-month follow-up. At study inclusion, plasma samples were prepared from whole blood following centrifugation for 20 min at 2000 g at room temperature and stored at −80˚C until further analyses. At the four-month follow-up, serum samples were prepared by centrifugation at 2000g for 10 minutes in room temperature and stored at -80˚C for further analyses. Serological analyses were performed as earlier described [29]. Briefly, a multiplex antigen bead array was used in high throughput 384-plates format using the FlexMap3D (Luminex Corp). IgG reactivity was measured towards spike trimers comprising the prefusion-stabilized spike glycoprotein ectodomain (in-house produced, expressed in HEK and purified using a C-terminal Strep II tag) and the C-terminal domain of the nucleocapsid protein (in-house produced, expressed in *Escherichia coli* and purified using a 427 C-terminal His-tag). The threshold for seropositive response for each protein was determined by the mean level plus six times the standard deviation from twelve negative controls analysed in each assay.

## Microneutralization assay

Microneutralization assay was performed as earlier described [30]. Briefly, serum was heat inactivated and 10-fold diluted in duplicate. Each dilution was mixed with tissue culture of SARS-CoV-2 and incubated. The cells were inspected for signs of cytopathogenic effect (CPE) by optical microscopy after four days. If <50% of the cell layer showed signs of CPE the well was scored as neutralizing.

## Statistical analyses

Antibody levels were compared using the Wilcoxon rank-sum test, and are presented as medians, interquartile ranges (IQR), and p-values. Age was compared using Student's t-test, and is presented as mean, standard deviation (SD), and p-value. Associations of categorical variables, *e.g.* serostatus, antibody persistence, symptoms, and sex, were examined using Fisher's exact test, and are presented as proportion, odds ratio (OR), and confidence interval (CI).

To assess the individual influence of anti-spike IgG levels and anti-nucleocapsid IgG levels on virus neutralization, these data were fitted in a logistic regression model specified as

$$\log(OR) = \beta_0 + \beta_1 MFI_{anti-spike\ IgG} + \beta_2 MFI_{anti-nucleocapsid\ IgG}$$
$$+ \beta_{12} MFI_{anti-spike\ IgG} MFI_{anti-nucleocapsid\ IgG}$$

where the product term was used to control for the interaction of anti-spike and anti-nucleocapsid IgGs. Model fit was estimated using the Cragg-Uhler pseudo-$R^2$, and odds ratios for an increase in MFI from 0 to 5000 AU were calculated using the estimated model coefficients as $e^{\beta \times 5000}$.

Statistics and data visualization were performed in R [31] using packages tidyverse, rlang, pander, knitr, scales, ggsignif, ggbeeswarm, exact2x2, egg, cowplot, jtools, and oddsratio.

## Acknowledgments

The authors are grateful to Carola Jonsson, Sofie Lundin, Camilla Redhevon, Sarah Juhlin, Nelly Romero, Anna Weimer, Jeanette Agge, Frida Holmström, Karina Halling, Tsige Mulugeta, Martha Kihlgren at Danderyd Hospital for assisting in administration and blood sampling. We thank Richard Scholvin for technical support with the smartphone app and assisting with data information, and Carina Rudberg and Christina Einarsson for assisting with data collection. The Protein Factory at KTH is acknowledged for protein production and purification and Sofia Bergström, Shaghayegh Bayati and Sara Mravinacova at KTH and SciLifeLab for technical assistance.

## Author Contributions

**Conceptualization:** Sebastian Havervall, Jonas Klingström, Sophia Hober, Peter Nilsson, Charlotte Thålin.

**Data curation:** Sebastian Havervall, August Jernbom Falk, Cecilia Hellström, Charlotte Thålin.

**Formal analysis:** Sebastian Havervall, August Jernbom Falk, Peter Nilsson, Charlotte Thålin.

**Funding acquisition:** Sophia Hober, Peter Nilsson, Charlotte Thålin.

**Investigation:** Sebastian Havervall, Jonas Klingström, Nina Greilert-Norin, Lena Gabrielsson, Ann-Christin Salomonsson, Eva Isaksson, Eni Andersson, Jennie Olofsson, Lovisa Skoglund, Jamil Yousef, Elisa Pin, Wanda Christ.

**Methodology:** Jonas Klingström, Cecilia Hellström, Elisa Pin, Mikaela Olausson, My Hedhammar, Hanna Tegel, Anna Månberg, Sophia Hober, Peter Nilsson.

**Project administration:** Sophia Hober, Peter Nilsson, Charlotte Thålin.

**Resources:** Jonas Klingström, Elisa Pin, Mikaela Olausson, My Hedhammar, Hanna Tegel, Anna Månberg, Sophia Hober, Peter Nilsson, Charlotte Thålin.

**Software:** August Jernbom Falk, Cecilia Hellström.

**Supervision:** Sophia Hober, Peter Nilsson, Charlotte Thålin.

**Visualization:** August Jernbom Falk.

**Writing – original draft:** Sebastian Havervall, Charlotte Thålin.

**Writing – review & editing:** Sebastian Havervall, August Jernbom Falk, Jonas Klingström, Henry Ng, Ann-Sofie Rudberg, Cecilia Hellström, Elisa Pin, Sara Mangsbo, Mia Phillipson, Anna Månberg, Sophia Hober, Peter Nilsson, Charlotte Thålin.

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
