## [Decision Letter · Decision Letter 0]

2 Nov 2021

PONE-D-21-27419SARS-CoV-2 induces a durable and antigen specific humoral immunity after asymptomatic to mild COVID-19 infectionPLOS ONE

Dear Dr. Thålin,

Thank you for submitting your manuscript to PLOS ONE. After careful consideration, we feel that it has merit but does not fully meet PLOS ONE’s publication criteria as it currently stands. Therefore, we invite you to submit a revised version of the manuscript that addresses the points raised during the review process.Please make rebuttal to the reviewers. 

We look forward to receiving your revised manuscript.

Kind regards,

Etsuro Ito

Academic Editor

PLOS ONE

Journal Requirements:

Reviewers' comments:

Reviewer's Responses to Questions

**Comments to the Author**

1. Is the manuscript technically sound, and do the data support the conclusions?

Reviewer #1: Partly

Reviewer #2: Yes

2. Has the statistical analysis been performed appropriately and rigorously? 

Reviewer #1: Yes

Reviewer #2: I Don't Know

3. Have the authors made all data underlying the findings in their manuscript fully available?

Reviewer #1: No

Reviewer #2: Yes

4. Is the manuscript presented in an intelligible fashion and written in standard English?

Reviewer #1: Yes

Reviewer #2: Yes

5. Review Comments to the Author

Reviewer #1: The authors previously published a similar study cited in reference 20 in September of 2020 showing health care workers (HCW) have a higher risk of contracting COVID-19. Here these authors essentially repeated the same serological assays and concluded quickly that this is a 4-month follow-up study to show anti-spike antibody titers were sustainable whereas anti-NP antibody titers were not.

I think the conclusion was very limited in the sense that this type of study reflected a research attitude of " salami slicing" which means that an tactic is used to provide fragmented data with a trade-off between quality and quantity of the researches. One may wonder that given the time frame now it has been more than one year since their previous report (ref 20) why these authors now report a follow-up study of only 4 months?

In addition, the conclusion brings in little new insight since nowadays more important question is to address cross-protection of antibody toward new emerging strains of virus of concern, such as delta variant and other questions alike.

Reviewer #2: The authors describe the serological responses to asymptomatic SARS-CoV-2 infection, indicating that, similar to symptomatic infection, antibodies against the S glycoprotein, including those with neutralising potential, remain at higher levels longer than antibodies against the nucleoprotein. Furthermore, the magnitude of antibody responses appeared to be lower in asymptomatic infection compared to symptomatic infection.

The objectives of the study are clearly described, the work is thorough, and the data are well well-described and support the conclusions drawn. Whilst these findings are not entirely novel, they do increase understanding of humoral responses in mild/asymptomatic infection. The authors do well to highlight that the asymptomatic and symptomatic cohorts differ widely, and so no direct comparisons can be made.

I like this study in its present form and have only a few minor suggestions.

1 – In the introduction, ‘Cross-sectional clinical, demographic and serological data of the HCW at study inclusion has been presented elsewhere [20]’. Would a brief description be appropriate here?

2 – in results, figure 3 – could the assay cut-offs be added to graphs to clearly depict positivity/negativity? One anti-S negative (orange) appears to have a higher anti-S IgG read out than some of the anti-S positives.

4 – The symbols and colours use in the figures could possibly be improved to make the data more accessible.

6. PLOS authors have the option to publish the peer review history of their article (what does this mean?). If published, this will include your full peer review and any attached files.

Reviewer #1: No

Reviewer #2: No

---

## [Author Response · Author response to Decision Letter 0]

26 Nov 2021

Reviewer #1: The authors previously published a similar study cited in reference 20 in September of 2020 showing health care workers (HCW) have a higher risk of contracting COVID-19. Here these authors essentially repeated the same serological assays and concluded quickly that this is a 4-month follow-up study to show anti-spike antibody titers were sustainable whereas anti-NP antibody titers were not.

I think the conclusion was very limited in the sense that this type of study reflected a research attitude of " salami slicing" which means that an tactic is used to provide fragmented data with a trade-off between quality and quantity of the researches. One may wonder that given the time frame now it has been more than one year since their previous report (ref 20) why these authors now report a follow-up study of only 4 months?

In addition, the conclusion brings in little new insight since nowadays more important question is to address cross-protection of antibody toward new emerging strains of virus of concern, such as delta variant and other questions alike.

We acknowledge that the results in the manuscripts are sub-analyses of a large study, and results have been published at several follow-ups (PMID: 34459525, PMID: 34391088, PMID: 33825846, PMID: 33033249). However, the data set is large and there are many aspects and findings worth reporting. Although the current manuscript was drafted several months ago, delays have prevented us from publishing until now. Nevertheless, we strongly believe the results are still very relevant considering the global spread of the virus. Published data constitute parts of the whole puzzle, and also data which corroborate findings form other studies are crucial. Furthermore, this manuscript focuses on the association between specific symptoms and disease severity, which is not within the scope of our other publications. The study period in this manuscript is also during a time with very limited spread in Sweden, reducing the effect of possible re-exposure on the association of antibody persistence and acute symptomatology. We also focus on the persistence of different specificities of anti-SARS-CoV-2 IgG in connection to symptoms and neutralization capacity, which has not been the focus at other follow-ups. We therefore believe that the data presented in this manuscript enhances and builds on data already published on this cohort, rather than corroborating already published findings.

Reviewer #2: The authors describe the serological responses to asymptomatic SARS-CoV-2 infection, indicating that, similar to symptomatic infection, antibodies against the S glycoprotein, including those with neutralising potential, remain at higher levels longer than antibodies against the nucleoprotein. Furthermore, the magnitude of antibody responses appeared to be lower in asymptomatic infection compared to symptomatic infection.

The objectives of the study are clearly described, the work is thorough, and the data are well well-described and support the conclusions drawn. Whilst these findings are not entirely novel, they do increase understanding of humoral responses in mild/asymptomatic infection. The authors do well to highlight that the asymptomatic and symptomatic cohorts differ widely, and so no direct comparisons can be made.

I like this study in its present form and have only a few minor suggestions.

1 – In the introduction, ‘Cross-sectional clinical, demographic and serological data of the HCW at study inclusion has been presented elsewhere [20]’. Would a brief description be appropriate here?

Thank you for this valuable suggestion. We have added a brief description there.

2 – in results, figure 3 – could the assay cut-offs be added to graphs to clearly depict positivity/negativity? One anti-S negative (orange) appears to have a higher anti-S IgG read out than some of the anti-S positives.

This is an important point. As described in the Methods, the cutoff is calculated as mean + 6sd of the negative controls on a per-assay-plate basis. This is done to minimize the effect of any inter-assay variations. Therefore, the cut-off will vary slightly between assay plates, and is not readily visualized as one single line. In our opinion, adding several cut-off lines in the plot would reduce rather than increase clarity.

4 – The symbols and colours use in the figures could possibly be improved to make the data more accessible.

We have improved the colors and shapes to improve readability and accessibility.

---

## [Editor Report · Decision Letter 1]

17 Dec 2021

SARS-CoV-2 induces a durable and antigen specific humoral immunity after asymptomatic to mild COVID-19 infection

PONE-D-21-27419R1

Dear Dr. Thålin,

We’re pleased to inform you that your manuscript has been judged scientifically suitable for publication and will be formally accepted for publication once it meets all outstanding technical requirements.

Kind regards,

Etsuro Ito

Academic Editor

PLOS ONE

---

## [Editor Report · Acceptance letter]

23 Dec 2021

PONE-D-21-27419R1 

SARS-CoV-2 induces a durable and antigen specific humoral immunity after asymptomatic to mild COVID-19 infection 

Dear Dr. Thålin:

I'm pleased to inform you that your manuscript has been deemed suitable for publication in PLOS ONE. Congratulations! Your manuscript is now with our production department. 

Kind regards, 

on behalf of

Prof. Etsuro Ito 

Academic Editor

PLOS ONE